# ERTACache: Error Rectification and Timesteps Adjustment for Efficient Diffusion

**Xurui Peng**[1][*], **Chenqian Yan**[1][*], **Hong Liu**[1][*],
**Rui Ma**[1], **Fangmin Chen**[1], **Xing Wang**[1], **Zhihua Wu**[1], **Songwei Liu**[1][‡], **Mingbao Lin**[2][†]
[1]ByteDance Inc. [2]Rakuten Asia

## Abstract

Diffusion models suffer from substantial computational overhead due to their inherently iterative inference process. While feature caching offers a promising acceleration strategy by reusing intermediate outputs across timesteps, naïve reuse often incurs noticeable quality degradation. In this work, we formally analyze the cumulative error introduced by caching and decompose it into two principal components: *feature shift error*, caused by inaccuracies in cached outputs, and *step amplification error*, which arises from error propagation under fixed timestep schedules. To address these issues, we propose **ERTACache**, a principled caching framework that jointly rectifies both error types. Our method employs an offline residual profiling stage to identify reusable steps, dynamically adjusts integration intervals via a trajectory-aware correction coefficient, and analytically approximates cache-induced errors through a closed-form residual linearization model. Together, these components enable accurate and efficient sampling under aggressive cache reuse. Extensive experiments across standard image and video generation benchmarks show that ERTACache achieves up to $2\times$ inference speedup while consistently preserving or even improving visual quality. Notably, on the state-of-the-art Wan2.1 video diffusion model, ERTACache delivers $2\times$ acceleration with minimal VBench degradation, effectively maintaining baseline fidelity while significantly improving efficiency.

## 1 Introduction

Diffusion models Ho et al. (2020); Song et al. (2020); Rombach et al. (2022); Labs (2024) have demonstrated remarkable capabilities, driving groundbreaking progress across diverse domains such as image Esser et al. (2024), video Wan et al. (2025), 3D content Kong et al. (2024); Huang et al. (2024a), and audio generation Schneider et al. (2023), largely owing to the scalability of transformer-based architectures Vaswani et al. (2017). To further improve generation quality, researchers have continually scaled up Diffusion Transformers (DiTs) Peebles & Xie (2023). However, this comes at the cost of significantly increased inference time and memory consumption—issues that are particularly pronounced in video generation, where synthesizing a 6-second 480p clip on an NVIDIA L20 GPU can take 2-5 minutes, posing serious limitations for practical deployment. To address these deficiencies, a variety of acceleration techniques have been proposed, including reduced-step samplers Lu et al. (2022); Liu et al. (2022), model distillation Zhou et al. (2024), quantization strategies Li et al. (2024), and cache-based acceleration Liu et al. (2025b;a).

Among various acceleration strategies, cache-based feature reuse has emerged as a particularly practical solution. By reusing intermediate model outputs across timesteps, this approach well reduces redundant computations during inference, yielding substantial speedups without requiring additional training overhead. Moreover, it generalizes readily to diverse video diffusion models, making it appealing for real-world deployment. However, existing cache-based methods still face key limitations. Approaches that cache internal transformer states Kahatapitiya et al. (2024); Ji et al. (2025) often incur high GPU memory costs. Meanwhile, methods like TeaCache Liu et al. (2025b), which

---

[*]Equal Contribution.

[†]Corresponding Authors.

[‡]Porject Leader.

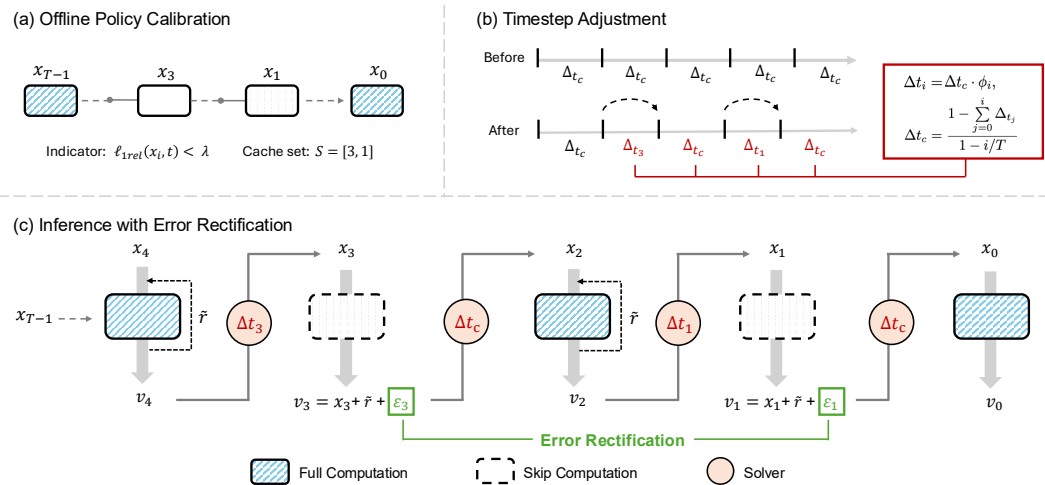

Figure 1: Framework of our proposed ERTACache.

choose to dynamically predict cache strategies based on input-dependent heuristics, can exhibit discrepancies between predicted reuse quality and actual reconstruction error, limiting reliability.

In this work, we begin with a systematic analysis of dynamic cache-based acceleration. Our empirical findings reveal that despite the input-dependent nature of diffusion trajectories, cache reuse patterns and associated errors exhibit strong consistency across prompts. This suggests that a generic, offline-optimized caching strategy may be sufficient to approximate optimal behavior in most scenarios. Furthermore, we dive into insights by decomposing the sources of cache-induced degradation and identifying two dominant error modes: (i) feature shift error, introduced by inaccuracies in the reused model outputs; and (ii) step amplification error, which arises from the temporal compounding of small errors due to fixed timestep schedules.

Motivated by the above insights, we formally introduce **ERTACache**, a principled framework for error-aware caching in diffusion models. Figure 1 gives an overview of our ERTACache framework. Our proposed ERTACache adopts a dual-dimensional correction strategy: (1) we first perform offline policy calibration by searching for a globally effective cache schedule using residual error profiling; (2) we then introduce a trajectory-aware timestep adjustment mechanism to mitigate integration drift caused by reused features; (3) finally, we propose an explicit error rectification that analytically approximates and rectifies the additive error introduced by cached outputs, enabling accurate reconstruction with negligible overhead.

Together, these components enable ERTACache to deliver high-quality generations while substantially reducing compute. Notably, our proposed ERTACache achieves over $50\%$ GPU computation reduction on video diffusion models, with visual fidelity nearly indistinguishable from full-computation baselines.

Our main contributions can be summarized as follows:

- We provide a formal decomposition of cache-induced errors in diffusion models, identifying two key sources: feature shift and step amplification.

- We propose ERTACache, a caching framework that integrates offline-optimized caching policies, timestep corrections, and closed-form residual rectification.

- Extensive experiments demonstrate that ERTACache consistently achieves over $2\times$ inference speedup on state-of-the-art video diffusion models such as Open-Sora 1.2, CogVideoX, and Wan2.1, with significantly better visual fidelity compared to prior caching methods.

## 2  RELATED WORK

**Diffusion-based Generation**   Early diffusion models such as DDPM Ho et al. (2020) produce high-resolution images through a 1000-step Markov denoising chain, but their sheer number of iterations incurs prohibitive latency. DDIM Song et al. (2020) later shortened this chain to 50-100 implicit steps, yet even these "fast" samplers remain slow for time-critical applications. More recent systems—*e.g.*, Stable Diffusion 3 Esser et al. (2024) with architectural streamlining and Flux-dev1.0 Labs (2024) with simplified Transformer blocks—lower the cost per step, but cannot escape the fundamental requirement of many denoising passes to preserve fidelity. Video generation magnifies the problem: adding a temporal dimension and higher spatial resolutions multiplies compute.

CogVideoX Yang et al. (2024), despite block-causal attention and a 3D-VAE compressor, still needs 113s (50 steps) to render a 480p six-frame clip on a single H800 GPU. OpenSora 1.2 Zheng et al. (2024) accelerates per-frame inference via a dual-branch design and dynamic sparse attention, while HunyuanVideo Kong et al. (2024) achieves a 1.75× speed-up with sliding-window attention and DraftAttention Shen et al. (2025a). WAN-2.1 Wan et al. (2025), which prioritizes temporal coherence, pushes latency even higher—200s for an 81-frame 480p clip on an A800 GPU—underscoring the gap to real-time use. The root cause is intrinsic: every denoising step forwards a large UNet Ronneberger et al. (2015) or Transformer whose parameter count scales roughly quadratically with input size. Consequently, despite impressive image and video quality, diffusion models remain hamstrung by extreme inference times, especially in the video domain.

**Cache-based Acceleration**   Because iterative denoising dominates runtime, feature caching has emerged as a promising accelerator. The idea is to exploit feature similarity across successive steps by storing and reusing intermediate activations. TeaCache Liu et al. (2025b) predicts layer outputs from inputs to decide what to cache, although its learned policy often collapses to a fixed schedule. Finer-grained schemes such as AdaCache Kahatapitiya et al. (2024), BAC Ji et al. (2025), and DiTFastAttn Yuan et al. (2024) cache blocks or attention maps, but their granularity causes memory to grow quadratically, leading to out-of-memory failures on long videos. Token-level approaches—including FastCache Liu et al. (2025a), TokenCache Lou et al. (2024), and Duca Zou et al. (2024)—are orthogonal to our method and complementary in principle. LazyDiT Shen et al. (2025b) imposes extra training losses to regularize cached features, incurring additional training overhead, whereas our strategy achieves quality retention via direct error compensation with no retraining. ICC Chen et al. (2025) adjusts cached features through low-rank calibration; in contrast, we explicitly model the residual error, ensuring both accuracy and memory efficiency during generation.

## 3  METHODOLOGY

### 3.1  PRELIMINARIES

**Diffusion Process**   Diffusion models generate data by gradually perturbing a real sample $x_0 \sim p_{data}$ into pure Gaussian noise $x_T \sim \mathcal{N}(0, I)$ and then inverting that transformation. Recent work—most notably Stable Diffusion 3 (SD3) Esser et al. (2024) and the Flow-Matching paradigm Lipman et al. (2022)—has positioned flow-based objectives as the dominant framework for both image and video synthesis. Under flow matching, the forward process is a simple linear interpolation:

$$x_t = (1 - t)x_0 + t \cdot x_T, \quad t \in [0, 1]. \tag{1}$$

whose trajectory has a constant true velocity field:

$$u_t = \frac{dx_t}{dt} = x_T - x_0. \tag{2}$$

A neural estimator $v_\theta(x_t, t)$ is trained to predict this velocity by minimizing the mean-squared error:

$$\mathcal{L}_{\text{FM}}(\theta) = \mathbb{E}_{t, x_0, x_T} \left[ \|v_\theta(x_t, t) - u_t\|_2^2 \right]. \tag{3}$$

During synthesis, one deterministically integrates the learned ODE from noise back to data. Discretizing the time axis for $i = T - 1, ..., 0$ yields:

$$x_{i-1} = x_i + \Delta t_i \cdot v_\theta(x_i, t), \tag{4}$$

where $\Delta t_i$ denotes the interval between two consecutive time steps. As $i \to 0$, the sample $x_0$ converges to the data distribution $p_{data}$.

**Residual Cache** Inference latency is dominated by repeated evaluations of the denoising network across many time steps. Residual caching tackles this bottleneck by exploiting the strong feature locality between successive steps: instead of storing full activations, we cache only the residual that the network adds to its input, preserving maximal information with minimal memory. Following prior work Chen et al. (2024); Liu et al. (2025b), we record at step $i + 1$:

$$\tilde{r} = v_\theta(x_{i+1}, t) - x_{i+1}. \tag{5}$$

If step $i$ is skipped, the model output is reconstructed on-the-fly as following:

$$\tilde{v}_i = x_i + \tilde{r}, \tag{6}$$

thereby avoiding a full forward pass while retaining high-fidelity features for the subsequent update.

## 3.2 ANALYZING ERROR ACCUMULATION FROM FEATURE CACHING

To understand the trade-offs in the cache-based acceleration scheme, we analyze how approximating model outputs with cached features introduces and accumulates error during the diffusion trajectory.

Let $\tilde{v}_i$ denote the cached model output at step $i$, approximating the true velocity $v_i \equiv v_\theta(x_i, t)$. This approximation introduces an additive error term $\varepsilon_i$:

$$\tilde{v}_i = v_i + \varepsilon_i. \tag{7}$$

Replacing $v_{i+1}$ with $\tilde{v}_{i+1}$ in Eq. (4) yields the cached latent:

$$\tilde{x}_i = x_{i+1} + \Delta t_{i+1} \cdot \tilde{v}_{i+1}. \tag{8}$$

Then, the deviation from the true trajectory after a single step is derived in the following:

$$\delta_i = \tilde{x}_i - x_i = \Delta t_{i+1} \cdot \varepsilon_{i+1}. \tag{9}$$

If the caching operation is performed for $m$ continuous steps right after the $i$-th step ($m \geq 1$) using Euler method, meaning that cached vectors replace actual computations during the $i$ to $i+m$ interval, then we have:

$$\tilde{x}_{i-m} = \tilde{x}_{i-m+1} + \Delta t_{i-m+1} \cdot \tilde{v}_{i-m+1}, \tag{10}$$

and the trajectory deviation $\delta_{t-m}$ compounds:

$$
\begin{aligned}
\delta_{i-m} &= \tilde{x}_{i-m} - x_{i-m} \\
&= \tilde{x}_{i-m+1} - x_{i-m+1} \\
&\quad + \Delta t_{i-m+1} \cdot \tilde{v}_{i-m+1} - \Delta t_{i-m+1} \cdot v_{i-m+1} \\
&= \delta_{i-m+1} + \Delta t_{i-m+1} \cdot \varepsilon_{t-m+1} \\
&= \cdots \\
&= \sum_{k=0}^{m-1} \underbrace{\Delta t_{i-k}}_{\substack{\text{Step Amplification} \\ \text{Error}}} \cdot \underbrace{\varepsilon_{i-k}}_{\substack{\text{Feature Shift} \\ \text{Error}}}.
\end{aligned}
\tag{11}
$$

This reveals two critical sources of error in cache-based acceleration: 1) **Feature Shift Error:** The discrepancy between cached features and ground-truth computations causes deviation in ODE trajectories. 2) **Step Amplification Error:** The pre-existing feature shift gets compounded through temporal integration, with error magnification.

## 3.3 ERTACACHE

Recent work such as TeaCache Liu et al. (2025b) has explored the potential of cache-based acceleration in diffusion sampling by predicting timestep-specific reuse strategies via online heuristics. Although effective to a certain extent, these methods still rely heavily on threshold tuning and often fail to generalize across different prompts or sampling trajectories. Motivated by our analysis that reveals consistent error patterns across diverse inputs (see Figure 2a), we propose a principled framework that combines *offline policy calibration*, *adaptive timestep adjustment*, and *explicit error rectification* to achieve more principled and reliable caching for diffusion models.

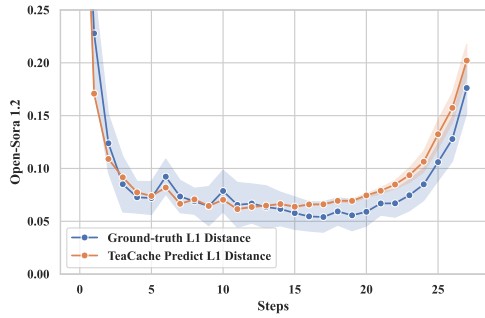
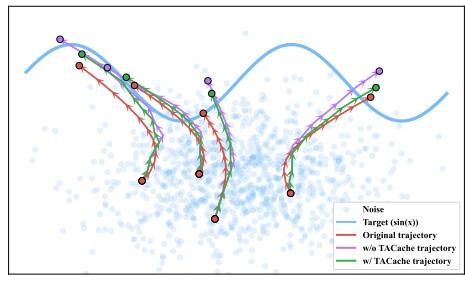

(a) Ground-truth vs. TeaCache's $\ell_1$ distance across timesteps

(b) ODE trajectories with/without timestep adjustment

Figure 2: (a) The ground-truth $\ell_1$ distance (blue) between real cached and computed features shows minor variation across timesteps. In contrast, Tea-Cache's predicted $\ell_1$ distance (orange) remains consistent across prompts but diverges significantly from ground-truth in later steps, indicating growing prediction error over time. (b) ODE trajectories with and without timestep adjustment.

**Offline Policy Calibration via Residual Error Profiling**  While previous approaches Liu et al. (2025b) predict caching decision on-the-fly, we instead formulate caching as an offline optimization problem. Let $r(x_i, t)$ denote the residual computed during standard inference. When caching is introduced, the residual to be reused is represented as $\tilde{r}$. Our strategy proceeds in three stages:

- **Ground-Truth Residual Logging.** We run full inference on a small calibration set and record ground-truth residuals $r^{gt}(x_i, t)$ for all steps.

- **Threshold-Based Policy Search.** For a range of candidate thresholds $\lambda$, we evaluate whether the cached residual $\tilde{r}^{cali}$ is sufficiently close to the freshly computed one using the relative $\ell_1$ error:

$$\ell_{1rel}(x_i, t) = \frac{\|\tilde{r}^{cali} - r^{cali}(x_i, t)\|_1}{\|r^{gt}(x_i, t)\|_1}. \tag{12}$$

  If $\ell_{1rel}(x_i, t) < \lambda$, the cache is reused; otherwise, the model recomputes and updates the cache. Sweeping across values of $\lambda$, we derive a reusable cached timestep set $S = \{s_0, s_1, \ldots, s_c\}$.

- **Inference-Time Cache Application.** During the inference stage, the model initializes with $v_{T-1}$ and caches the residual $\tilde{r} = v_{T-1} - x_{T-1}$. Then, for each step $t$, if $t \in S$, it reuses $\tilde{r}$; otherwise, it computes $v_t$ from scratch and refreshes the cache $\tilde{r}$.

The choice of $\lambda$ involves a trade-off: lower values preserve finer details through frequent cache updates, whereas higher values prioritize generation speed at the potential cost of visual fidelity. Optimal threshold selection should balance these competing objectives.

**Trajectory-Aware Timestep Adjustment**  A naïve reuse of cached features assumes strict adherence to the original ODE trajectory. However, as visualized in Figure 2b, this leads to noticeable trajectory drift, especially when fixed timestep intervals are maintained. To counteract this deviation, we introduce a trajectory-aware timestep adjustment mechanism, where a correction coefficient $\phi_t \in [0, 1]$ dynamically modifies the timestep size.

- Initially, $\Delta t_c = 1/T$ for uniform sampling.

- For each step $i$, we apply:

$$\Delta t_i = \begin{cases} \Delta t_c, & \text{if } i \notin S, \\ \Delta t_c \cdot \phi_i, & \text{if } i \in S, \end{cases} \tag{13}$$

  with $\phi_i = \text{clip}\left(1 - \frac{\|\tilde{v}_i - v_i\|_1}{\|v_i - v_{i+1}\|_1}, \ 0, \ 1\right).$

- After each update, the residual timestep budget is adjusted by following:

$$\Delta t_c = \frac{1 - \sum_{j=0}^{i} \Delta_{t_j}}{1 - i/T}. \tag{14}$$

This adaptive policy helps align the actual trajectory with the intended sampling path, even under aggressive reuse. The simplified update workflow is depicted in Figure 1(b).

**Explicit Error Rectification via Residual Linearization**  To further mitigate error accumulation, we introduce an explicit error modeling component. Since the additive error $\varepsilon_i = \tilde{v}_i - v_i$ is difficult to predict directly due to prompt-specific structure, we approximate it using a lightweight linearized model:

$$\varepsilon_i = \sigma(K_i \cdot \tilde{v}_i + B_i), \tag{15}$$

where $\sigma$ is sigmoid, and $n = N_t \cdot N_c \cdot N_h \cdot N_w$, where $N_t$, $N_c$, $N_h$ denote dimension of time, height and width for latents.

The mean squared error loss is defined as:

$$L(K_i) = \frac{1}{n} \sum_{j=1}^{n} [\varepsilon_{ij} - \sigma(K_{ij}\tilde{v}_{ij} + B_{ij})]^2. \tag{16}$$

Using a first-order Taylor approximation of the sigmoid:

$$\sigma(K_i\tilde{v}_i + B_i) \approx \frac{1}{4}(K_i\tilde{v}_i + B_i) + \frac{1}{2}. \tag{17}$$

Substituting into the loss function and computing the partial derivatives *w.r.t.* $B_i$ and $K_i$, we obtain:

$$\frac{\partial L}{\partial B_i} = -\frac{1}{2n} \sum_{j=1}^{n} [\varepsilon_{ij} - \frac{1}{4}(K_{ij}\tilde{v}_{ij} + B_{ij}) - \frac{1}{2}] = 0. \tag{18}$$

$$\frac{\partial L}{\partial K_i} = -\frac{1}{2n} \sum_{j=1}^{n} \tilde{v}_{ij}[\varepsilon_{ij} - \frac{1}{4}(K_{ij}\tilde{v}_{ij} + B_{ij}) - \frac{1}{2}] = 0. \tag{19}$$

Let:

$$\begin{cases} \bar{\varepsilon}_i = \frac{1}{n} \sum_{j=1}^{n} \varepsilon_{ij}, \\ \bar{v}_i = \frac{1}{n} \sum_{j=1}^{n} \tilde{v}_{ij}, \\ S_{v_i v_i} = \sum_{j=1}^{n} (\tilde{v}_{ij} - \bar{v}_i)^2, \\ S_{v_i \varepsilon_i} = \sum_{j=1}^{n} (\tilde{v}_i - \bar{v}_i)(\varepsilon_{ij} - \bar{\varepsilon}_i). \end{cases} \tag{20}$$

An approximate closed-form solution can be derived as follows (see Appendix for detailed derivation):

$$K_i \approx 4 \cdot \frac{S_{v_i \varepsilon_i}}{S_{v_i v_i}}, \quad B_i \approx 4(\bar{\varepsilon}_i - \frac{1}{2}) - 4K_i\bar{v}_i. \tag{21}$$

These values can be precomputed from a small extracted dataset and reused during inference to provide error-corrected cached outputs.

## 4 EXPERIMENTS

### 4.1 EXPERIMENTAL SETUP

**Base Models and Compared Methods**  To substantiate the generality and efficacy of our proposed ERTACache, we integrate it into state-of-the-art diffusion backbones spanning both video and image generation. Concretely, we evaluate Open-Sora 1.2 Zheng et al. (2024), CogVideoX Yang et al. (2024), Wan2.1 Wan et al. (2025), and Flux-dev 1.0 Labs (2024). We benchmark these models against leading training-free acceleration techniques—namely, PAB Zhao et al., $\Delta$-DiT Chen et al. (2024), FasterCache Lv et al. (2024), ProfilingDiT Ma et al. (2025), and TeaCache Liu et al. (2025b)—to isolate and quantify the incremental gains conferred by our approach.

| Method | Efficiency | | Visual Quality | | | |
|---|---|---|---|---|---|---|
| | Speedup↑ | Latency (s)↓ | VBench ↑ | LPIPS↓ | SSIM ↑ | PSNR ↑ |
| **Open-Sora 1.2** (51 frames, 480P) | | | | | | |
| Open-Sora 1.2 ($T = 30$) | 1× | 44.56 | 79.22% | - | - | - |
| Δ-DiT Chen et al. (2024) | 1.03× | - | 78.21% | 0.5692 | 0.4811 | 11.91 |
| T-GATE Liu et al. (2025c) | 1.19× | - | 77.61% | 0.3495 | 0.6760 | 15.50 |
| PAB-slow Zhao et al. | 1.33× | 33.40 | 77.64% | 0.1471 | 0.8405 | 24.50 |
| PAB-fast Zhao et al. | 1.40× | 31.85 | 76.95% | 0.1743 | 0.8220 | 23.58 |
| TeaCache-slow Liu et al. (2025b) | 1.55× | 28.78 | 79.28% | 0.1316 | 0.8415 | 23.62 |
| TeaCache-fast Liu et al. (2025b) | 2.25× | 19.84 | 78.48% | 0.2511 | 0.7477 | 19.10 |
| **ERTACache-slow** ($\lambda$=0.1) | **1.55×** | **28.75** | **79.36%** | **0.1006** | **0.8706** | **25.45** |
| **ERTACache-fast** ($\lambda$=0.18) | **2.47×** | **18.04** | **78.64%** | **0.1659** | **0.8170** | **22.34** |
| **CogVideoX** (48 frames, 480P) | | | | | | |
| CogVideoX ($T = 50$) | 1× | 78.48 | 80.18% | - | - | - |
| Δ-DiT ($N_c = 4, N = 2$) | 1.08× | 72.72 | 79.61% | 0.3319 | 0.6612 | 17.93 |
| Δ-DiT ($N_c = 8, N = 2$) | 1.15× | 68.19 | 79.31% | 0.3822 | 0.6277 | 16.69 |
| Δ-DiT ($N_c = 12, N = 2$) | 1.26× | 62.50 | 79.09% | 0.4053 | 0.6126 | 16.15 |
| PAB Zhao et al. | 1.35× | 57.98 | 79.76% | 0.0860 | 0.8978 | 28.04 |
| FasterCache Lv et al. (2024) | 1.62× | 48.44 | 79.83% | 0.0766 | 0.9066 | 28.93 |
| TeaCache Liu et al. (2025b) | 2.92× | 26.88 | 79.00% | 0.2057 | 0.7614 | 20.97 |
| **ERTACache-slow** ($\lambda$=0.08) | **1.62×** | **48.44** | **79.30%** | **0.0368** | **0.9394** | **32.77** |
| **ERTACache-fast** ($\lambda$=0.3) | **2.93×** | **26.78** | **78.79%** | **0.1012** | **0.8702** | **26.44** |
| **Wan2.1-1.3B** (81 frames, 480P) | | | | | | |
| Wan2.1-1.3B ($T = 50$) | 1× | 199 | 81.30% | - | - | - |
| TeaCache Liu et al. (2025b) | 2.00× | 99.5 | 76.04% | 0.2913 | 0.5685 | 16.17 |
| ProfilingDiT Ma et al. (2025) | 2.01× | 99 | 76.15% | 0.1256 | 0.7899 | 22.02 |
| **ERTACache** ($\lambda$=0.08) | **2.17×** | **91.7** | **80.73%** | **0.1095** | **0.8200** | **23.77** |

Table 1: Quantitative comparison in video generation models. ↑: higher is better; ↓: lower is better.

| Method | Efficiency | | Visual Quality | | | |
|---|---|---|---|---|---|---|
| | Speedup↑ | Latency (s)↓ | CLIP ↑ | LPIPS↓ | SSIM ↑ | PSNR ↑ |
| Flux-dev 1.0 ($T = 30$) | 1× | 26.14 | - | - | - | - |
| TeaCache Liu et al. (2025b) | 1.84× | 14.21 | 0.9065 | 0.4427 | 0.7445 | 16.4771 |
| **ERTACache** ($\lambda = 0.6$) | **1.86×** | **14.01** | **0.9534** | **0.3029** | **0.8962** | **20.5146** |

Table 2: Quantitative comparison in Flux-dev 1.0. ↑: higher is better; ↓: lower is better.

**Evaluation Metrics** We assess both video and image generation accelerators along two orthogonal axes: computational efficiency and perceptual fidelity.

- **Efficiency.** Inference latency—measured end-to-end from prompt ingestion to final frame—is reported in speedup compared to the base models.
- **Visual Quality.** We adopt a multi-faceted protocol that combines a human-centric benchmark with established similarity metrics, including: VBench Huang et al. (2024b), LPIPS Zhang et al. (2018), PSNR, SSIM and CLIP Hessel et al. (2021).

Together, these metrics yield a holistic view of how aggressively an acceleration method trades computational speed for generative fidelity.

**Implementation Details** For the primary quantitative analysis on the VBench default prompts, we synthesize five videos per prompt with distinct random seeds on NVIDIA A800 40 GB GPUs. To guarantee reproducibility, prompt extension is disabled and all other hyper-parameters are set to the official defaults released by each baseline. This protocol mirrors the evaluation pipelinesof recent studies Liu et al. (2025b); Zhao et al., ensuring a rigorously fair comparison of generation quality. To accelerate ablation studies, we further adopt a lightweight configuration in which Wan2.1-1.3B generates a single video per prompt with seed 0.

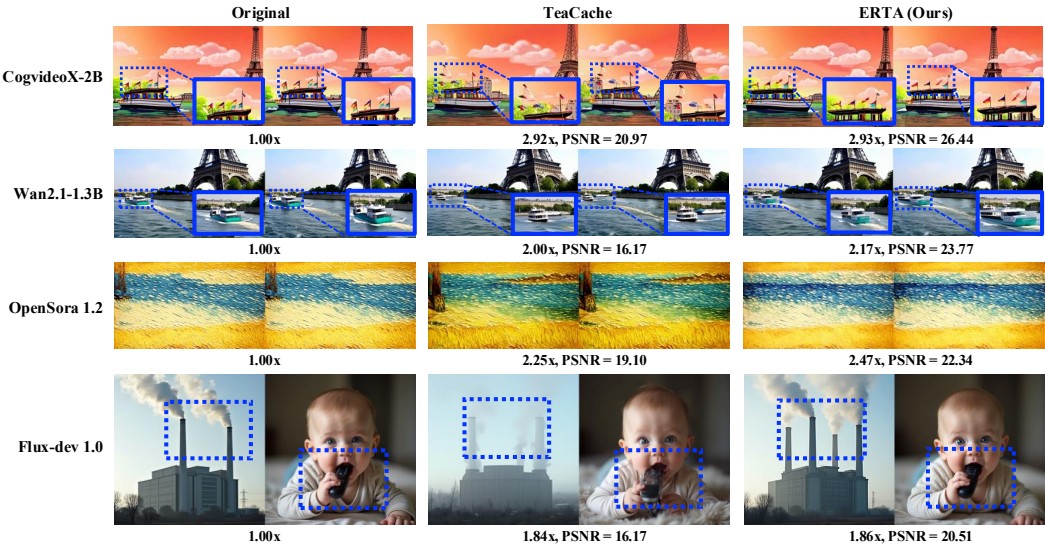

Figure 3: Comparison of visual quality and computational efficiency against competing approaches, illustrated by the first and last frames of generated video sequences.

## 4.2 MAIN RESULTS

**Quantitative Comparison** Table 1 and Table 2 present a quantitative evaluation on VBench, comparing efficiency and visual quality across baseline models. We assess two variants of ERTACache: a slow version (lower $\lambda$) for higher quality, and a fast version (higher $\lambda$) for greater speedup. Both variants consistently achieve strong acceleration and superior visual fidelity across diverse video and image synthesis models. On Open-Sora 1.2 Zheng et al. (2024), ERTACache-slow and ERTACache-fast yield speedups of $1.55\times$ and $2.47\times$, respectively, while outperforming $\Delta$-DiT, PAB, and Tea-Cache on all quality metrics. For CogVideoX, ERTACache achieves significant improvements in LPIPS (**0.1012**↓), SSIM (**0.8702**↑), and PSNR (**26.44**↑) compared to TeaCache (0.2057, 0.7614, 20.97), with a comparable $2.93\times$ speedup. With Wan2.1, we attain the best visual quality and a competitive $2.17\times$ speedup. ERTACache also demonstrates strong performance on the Flux-dev 1.0 image synthesis model, achieving the highest visual quality with a $1.86\times$ speedup, matching TeaCache.

**Visualization** The core strength of ERTACache lies in its ability to maintain high perceptual fidelity while ensuring strong temporal coherence. In Figure 3, ERTACache preserves fine-grained visual details and frame-to-frame consistency, outperforming TeaCache Liu et al. (2025b) and matching the non-cache reference. In video generation tasks using CogVideoX, Wan2.1-1.3B, and Opera-Sora, ERTACache achieves noticeably better temporal consistency, particularly between the first and last frames. When applied to the Flux-dev 1.0 image model, it enhances visual richness and detail. These results highlight ERTACache as a uniquely effective solution that balances visual quality and computational efficiency for consistent video generation.

## 4.3 ABLATION STUDY

We perform ablation studies on Wan2.1 and Flux-dev 1.0 to evaluate the individual impact of the offline-searched policy, timestep adjustment, and error rectification, as shown in Table 3. These results validate the effectiveness and efficiency of each component in our framework. We assess the sensitivity of ERTACache to varying data volumes, validating its robustness with a minimal computational overhead of less than 0.5%. Further details regarding this analysis are provided in the Appendix.

**Advantages of Offline Policy** Compared to *Uniform Cache*, the proposed *Offline Policy* effectively mitigates information loss, leading to consistent improvements across all evaluated metrics on both video and image synthesis models. It is noteworthy that the *Offline Policy* yields a statistically

| Wan2.1-1.3B | VBench↑ | LPIPS↓ | SSIM↑ | PSNR↑ |
|---|---|---|---|---|
| Uniform Cache | 79.35% | 0.5041 | 0.4058 | 13.7640 |
| Offline Policy | 80.59% | 0.1477 | 0.7738 | 22.0920 |
| +Time Adjustment | 80.89% | 0.1267 | 0.7988 | 22.9413 |
| +Error Rectification | 80.73% | 0.1095 | 0.8200 | 23.7680 |
| **Flux-dev 1.0** | **CLIP↑** | **LPIPS↓** | **SSIM↑** | **PSNR↑** |
| Uniform Cache | 0.9066 | 0.4424 | 0.7105 | 15.9428 |
| Offline Policy | 0.9433 | 0.3469 | 0.8693 | 19.4069 |
| +Time Adjustment | 0.9510 | 0.3268 | 0.8921 | 20.3586 |
| +Error Rectification | 0.9534 | 0.3029 | 0.8962 | 20.5146 |

Table 3: Ablation study on Wan2.1-1.3B and Flux-dev 1.0.

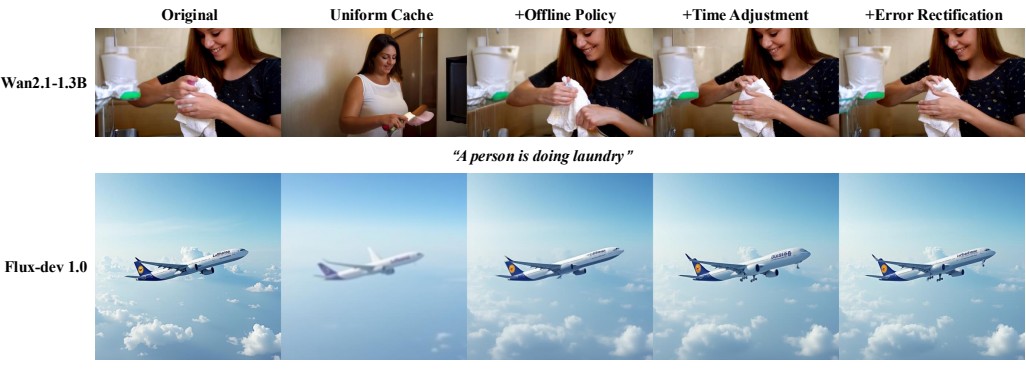

Figure 4: Visualization effects of each strategy in ERTACache.

significant improvement of 1.24% in the composite VBench metric on the Wan2.1-1.3B, indicating substantial enhancements in temporal consistency, motion authenticity, and visual aesthetics.

**Advantages of Timestep Adjustment**  The *Time Adjustment* mechanism enhances structural coherence, yielding significant improvements in evaluation metrics. For instance, on the Flux-dev 1.0, integrating this adjustment with the *Offline Policy* improves PSNR and SSIM by 0.9517 and 0.0228, respectively, compared to using the *Offline Policy* alone.

**Advantages of Error Rectification**  *Error Rectification* further improves perceptual fidelity by effectively correcting generation artifacts. As demonstrated in Figure 4, this module significantly enhances output quality through systematic error mitigation. Cumulatively, the full pipeline demonstrated notable gains in CLIP, LPIPS and PSNR versus baseline. This hierarchical progression demonstrates complementary component effects: offline policy enables foundational recovery, time adjustment optimizes temporal consistency, and error rectification maximizes preservation of detail, collectively resolving key efficiency-quality tradeoffs.

## 5 CONCLUSION

In this work, we present **ERTACache**, a principled and efficient caching framework for accelerating diffusion model inference. By decomposing cache-induced degradation into feature shift and step amplification errors, we develop a dual-path correction strategy that combines offline-calibrated reuse scheduling, trajectory-aware timestep adjustment, and closed-form residual rectification. Unlike prior heuristics-based methods, ERTACache provides a theoretically grounded yet lightweight solution that significantly reduces redundant computations while maintaining high-fidelity outputs. Empirical results across multiple benchmarks validate its effectiveness and generality, highlighting its potential as a practical solution for efficient generative sampling.

# 6 ETHICS STATEMENT

This work adheres to the ICLR Code of Ethics. In this study, no human subjects or animal experimentation was involved. All datasets used, including VBench, were sourced in compliance with relevant usage guidelines, ensuring no violation of privacy. We have taken care to avoid any biases or discriminatory outcomes in our research process. No personally identifiable information was used, and no experiments were conducted that could raise privacy or security concerns. We are committed to maintaining transparency and integrity throughout the research process.

# 7 REPRODUCIBILITY STATEMENT

We have made every effort to ensure that the results presented in this paper are reproducible. All code and datasets have been made publicly available in an anonymous repository to facilitate replication and verification. The experimental setup, including training steps, model configurations, and hardware details, is described in detail in the paper. We have also provided a full description of the evaluation metrics, hyperparameter settings, and any preprocessing steps applied to the data, to assist others in reproducing our experiments. We believe these measures will enable other researchers to reproduce our work and further advance the field.

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

## A APPENDIX

### A.1 ALGORITHM

In this section, we elaborate on the inference algorithm of ERTACache, encompassing the implementation of cache inference using Timestep Adjustment and Error Rectification methods (Algorithm 1), as well as the acquisition of a static cache list through Offline Policy Calibration (Algorithm 2).

---

**Algorithm 1** ERTACache Inference with Timestep Adjustment and Error Rectification

---

1: **Input:** Transformer model $M$, total step $T$, cached timesteps set $S$, Fixed parameter $K$, $B$
2: **Output:** Final output $x_0$
    *// Trajectory-Aware Timestep Adjustment*
3: Initialize $\Delta t_c = 1/T$
4: **for** $i = T - 1, ..., 0$ **do**
5:     **if** $i \in S$ **then**
6:         $\Delta t_i = \Delta t_c \cdot \phi_i$
7:         $\phi_i = \text{clip}\left(1 - \frac{\|\tilde{v}_i - v_i\|_1}{\|v_i - v_{i+1}\|_1},\ 0,\ 1\right)$
8:         $\Delta t_c = \frac{1 - \sum\limits_{j=1}^{i} \Delta t_j}{1 - i/T}$
9:     **else**
10:        $\Delta t_i = \Delta t_c$
11:     **end if**
12: **end for**
    *// ERTACache Inference*
13: Sample $x_{T-1} \sim \mathcal{N}(0, \mathbf{I})$, $i \leftarrow T - 1, t \in [0, 1]$
14: **for** $i = T - 1, ..., 0$ **do**
15:     **if** $i < T - 1$ **then**
16:         **if** $i \in S$ **then**
17:             $x_i = x_{i+1} + \Delta t_{i+1} \cdot (v_{i+1} - \sigma(K_{i+1} v_{i+1} + B_{i+1}))$
18:         **else**
19:             $x_i = x_{i+1} + \Delta t_{i+1} \cdot v_{i+1}$
20:         **end if**
21:     **end if**
22:     **if** $i \in S$ **then**
23:         $\tilde{v}_i = x_i + \tilde{r}$
24:     **else**
25:         $v_i \leftarrow$ compute output of the $M$
26:         $\tilde{r} = v_i - x_i$
27:     **end if**
28: **end for**

---

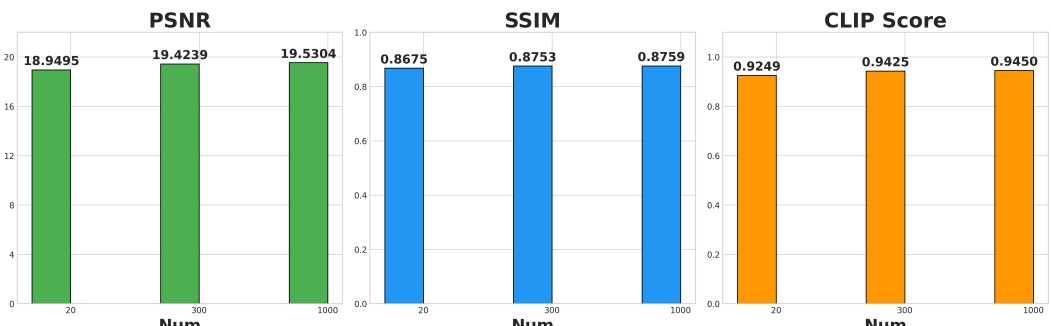

Figure 5: Illustration of different metrics using different number of prompts with timestep adjustment.

---

**Algorithm 2** Offline Policy Calibration via Residual Error Profiling

---

1: **Input:** Transformer model $M$, batched calibrate prompts set $P_c$, total step $T$, threshold $\lambda$
2: **Output:** Cached timesteps set $S$
3: Sample $x_{T-1} \sim \mathcal{N}(0, \mathbf{I})$, $i \leftarrow T-1$, $t \in [0, 1]$
   // *Ground-Truth Residual Logging*
4: **for** $i = T-1, ..., 0$ **do**
5:     $v_i \leftarrow$ compute output of the $M$ for $P_c$
6:     $r^{gt}(x_i, t) = v_i - x_i$
7: **end for**
   // *Threshold-Based Policy Search*
8: **for** $i = T-1, ..., 0$ **do**
9:     $v_i \leftarrow$ compute output of the $M$
10:    $r^{cali}(x_i, t) = v_i - x_i$
11:    **if** $i == T-1$ *or* $i == 0$ **then**
12:        $\tilde{r}^{cali} = v_i - x_i$
13:    **else**
14:        $\ell_{1rel}(x_i, t) = \frac{\|\tilde{r}^{cali} - r^{cali}(x_i,t)\|_1}{\|r^{gt}(x_i,t)\|_1}$
15:        **if** $\ell_{1rel} < \lambda$ **then**
16:            put $i$ to cached timesteps set $S$
17:            $\tilde{v_i} = x_i + \tilde{r}^{cali}$
18:        **else**
19:            $\tilde{r}^{cali} = v_i - x_i$
20:        **end if**
21:    **end if**
22: **end for**

---

| Model | Method | Memory Usage (MiB) | Latency (s) |
|---|---|---|---|
| FLUX-dev 1.0 | Baseline | 26988 | 26.14 |
| | TeaCache Liu et al. (2025b) | 27056 | 14.21 |
| | ERTACache Offline Policy (+Time Adjustment) | 27020 | 13.41 |
| | ERTACache Offline Policy (+Error Rectification) | 27022 | 14.01 |
| OpenSora 1.2 | Baseline | 21558 | 44.56 |
| | TeaCache Liu et al. (2025b) | 21558 | 19.84 |
| | ERTACache Offline Policy (+Time Adjustment) | 21346 | 17.79 |
| | ERTACache Offline Policy (+Error Rectification) | 21346 | 18.04 |

Table 4: Memory Usage and Latency on a single NVIDIA A800 40GB GPU.

## A.2 ANALYSIS OF ERROR RECTIFICATION IMPACT

**Sensitivity Analysis** In the Error Rectification process, we found that averaging the cache errors collected at each step under different prompts can improve efficiency and optimize model performance compared with determining the $K$ value through model training. We explore the impact of the number of prompts on the final results to evaluate the generality and accuracy of the averaging method. As shown in the Figure 5, the obtained different K values all exhibit stable trends in various metrics (PSNR, SSIM, CLIP Score) on the same medium-sized test dataset (data volume = 1000, based on MS COCO 2014 evaluation dataset) under different numbers of prompts (num = 20, 30, 1000). The experimental results demonstrate that the averaging method has strong stability and generality, and it can generalize to correction tasks in other scenarios with a very small batch of real errors, thus possessing the advantages of low cost and high efficiency.

**Extra Computational Cost** This study compares memory usage among three methods: No cache added, TeaCache Liu et al. (2025b), and our proposed ERTACache. The experimental results demonstrate that ERTACache exhibits a slight advantage in terms of memory usage (see Table 4). The core reason lies in the fact that ERTACache adopts an offline method to pre-determine a fixed cache list,

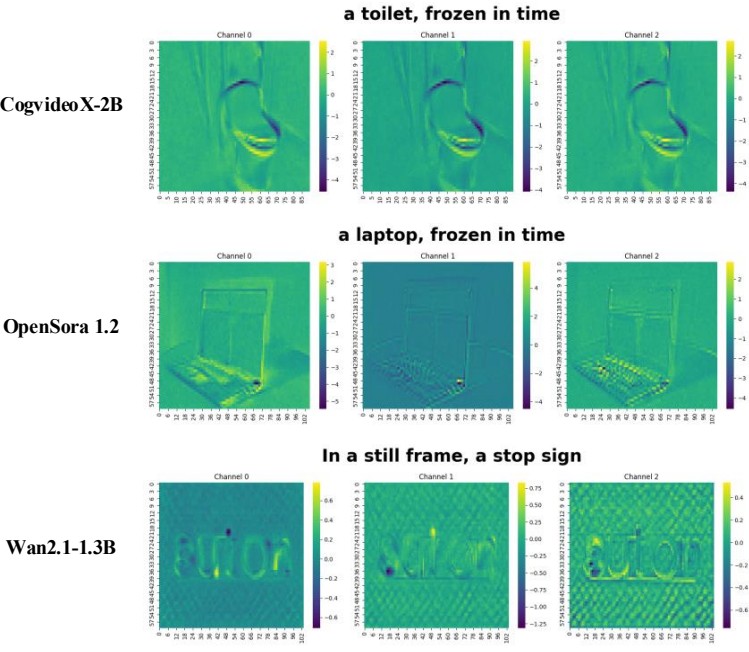

Figure 6: Illustration of cache error using different model, displaying 3 different channels of the first frame

thus eliminating the need for additional computation of modulated inputs and prediction of whether caching is required at the beginning of each step, thereby avoiding extra runtime computational overhead. Additionally, evaluating the Error Rectification parameter K reveals negligible impacts on both memory usage (FLUX-dev 1.0: +0.007%; OpenSora 1.2: +0%) and latency (FLUX-dev 1.0: +0.45%; OpenSora 1.2: +0.14%) across text-to-image and text-to-video scenarios.

### A.3 ERROR RECTIFICATION DERIVATION

The general formula for the accumulates error during the diffusion trajectory is:

$$\delta_{i-m} = \sum_{k=0}^{m-1} \Delta t_{i-k} \varepsilon_{i-k},$$

(22)

where $\varepsilon_i$ is an unknown cache error at $i$-th step. In past experiments, it has been found that each step of $\varepsilon_i$ contains structural information based on different prompts (see Figure 6), which has become a difficulty in the general error formula. Therefore, an attempt is made to combine the model cached output to make a generalized prediction for $\varepsilon_i$. For different prompts, it is assumed that there is a fixed high - dimensional tensor $K_i$, $B_i$ and an activation function $\sigma$, such that:

$$\varepsilon_i = \tilde{v}_\theta(x_i, t) - \mu_\theta(x_i, t) \approx \sigma(K_i * \tilde{v}_\theta(x_i, t) + B_i),$$

(23)

$$\sigma(x) = \frac{1}{1 + e^{-x}},$$

(24)

where $\tilde{v}_\theta(x_i, t)$ is the model cached output at $i$-th step, and $\mu_\theta(x_i, t)$ is the original model output at $i$-th step, $\sigma(x)$ is selected as the Sigmoid function. This method is equivalent to defining a single-layer convolutional network mapping structure, which can better accommodate the structural information brought by different prompts and make $\varepsilon_i$ universal.

Let the approximate model $A_i = \sigma(K_i \tilde{v}_i + B_i)$. We utilize the mean squared error as the loss function to minimize the error between predicted outputs $A_i$ and ground-truth values $\varepsilon_i$, where $n = N_t \times N_c \times N_h \times N_w$:

$$L(K_i) = \sum_{t=1}^{N_t} \sum_{c=1}^{N_c} \sum_{h=1}^{N_h} \sum_{w=1}^{N_w} (\varepsilon_{i,t,c,h,w} - A_{i,t,c,h,w})^2 = \frac{1}{n} \sum_{j=1}^{n} [\varepsilon_{ij} - \sigma(K_{ij}\tilde{v}_{ij} + B_{ij})]^2.$$

(25)

**Derivative with respect to $K_i$ and $B_i$**

$$\frac{\partial L}{\partial B_i} = \frac{2}{n}\sum_{j=1}^{n}[\varepsilon_{ij} - \sigma(K_{ij}\tilde{v}_{ij} + B_{ij})]\cdot\sigma(K_{ij}\tilde{v}_{ij} + B_{ij})[1 - \sigma(K_{ij}\tilde{v}_{ij} + B_{ij})]\cdot(-1) = 0. \quad (26)$$

$$\frac{\partial L}{\partial K_i} = \frac{2}{n}\sum_{j=1}^{n}[\varepsilon_{ij} - \sigma(K_i\tilde{v}_{ij} + B_{ij})]\cdot\sigma(K_{ij}\tilde{v}_{ij} + B_{ij})[1 - \sigma(K_{ij}\tilde{v}_{ij} + B_{ij})]\cdot(-\tilde{v}_{ij}) = 0. \quad (27)$$

Since the Sigmoid function is a nonlinear equation and the system of equations contains the coupling terms of $K$ and $B$ at the same time, the above system of equations cannot directly find an analytical solution. The Taylor expansion method is used for approximate linear processing:

$$\sigma(x) = \frac{1}{1 + e^{-x}}$$
$$= \frac{1}{2} + \frac{1}{4}x - \frac{1}{48}x^3 + \cdots + O(x^8).$$

Then we have:

$$A_i \approx \frac{1}{4}(K_i\tilde{v}_i + B_i) + \frac{1}{2}. \quad (28)$$

Substituting into the loss function, we derive:

$$L_i \approx \frac{1}{n}\sum_{j=1}^{n}\left[\varepsilon_{ij} - \frac{1}{4}(K_{ij}\tilde{v}_{ij} + B_{ij}) - \frac{1}{2}\right]^2. \quad (29)$$

**Derivative with respect to $K_i$ and $B_i$**

$$\frac{\partial L}{\partial B_i} = -\frac{1}{2n}\sum_{j=1}^{n}\left[\varepsilon_{ij} - \frac{1}{4}(K_{ij}\tilde{v}_{ij} + B_{ij}) - \frac{1}{2}\right] = 0. \quad (30)$$

$$\frac{\partial L}{\partial K_i} = -\frac{1}{2n}\sum_{j=1}^{n}\tilde{v}_{ij}\left[\varepsilon_{ij} - \frac{1}{4}(K_{ij}\tilde{v}_{ij} + B_{ij}) - \frac{1}{2}\right] = 0. \quad (31)$$

LINEAR SYSTEM OF EQUATIONS AND SOLUTION

After arrangement, a linear system of equations is obtained:

$$\begin{cases} \sum_{j=1}^{n}\left(\varepsilon_{ij} - \frac{1}{4}K_{ij}\tilde{v}_{ij} - \frac{1}{4}B_{ij} - \frac{1}{2}\right) = 0, \\ \sum_{j=1}^{n}\tilde{v}_{ij}\left(\varepsilon_{ij} - \frac{1}{4}K_{ij}\tilde{v}_{ij} - \frac{1}{4}B_{ij} - \frac{1}{2}\right) = 0. \end{cases} \quad (32)$$

It is further simplified to:

$$\begin{cases} n\left(\frac{1}{4}B_i + \frac{1}{2}\right) + \frac{1}{4}K_i\sum_{j=1}^{n}\tilde{v}_{ij} = \sum_{j=1}^{n}\varepsilon_{ij}, \\ \frac{1}{4}B_i\sum_{j=1}^{n}\tilde{v}_{ij} + \frac{1}{4}K_i\sum_{j=1}^{n}\tilde{v}_{ij}^2 + \frac{1}{2}\sum_{j=1}^{n}\tilde{v}_{ij} = \sum_{j=1}^{n}\tilde{v}_{ij}\varepsilon_{ij}. \end{cases} \quad (33)$$

Let:

$$\begin{cases} \bar{\varepsilon}_i = \frac{1}{n}\sum_{j=1}^{n}\varepsilon_{ij}, \\ \bar{v}_i = \frac{1}{n}\sum_{j=1}^{n}\tilde{v}_{ij}, \\ S_{v_iv_i} = \sum_{j=1}^{n}(\tilde{v}_{ij} - \bar{v}_i)^2 = \sum_{j=1}^{n}\tilde{v}_{ij}^2 - n\bar{v}_i^2, \\ S_{v_i\varepsilon_i} = \sum_{j=1}^{n}(\tilde{v}_{ij} - \bar{v}_i)(\varepsilon_{ij} - \bar{\varepsilon}_i) = \sum_{j=1}^{n}\tilde{v}_{ij}\varepsilon_{ij} - n\bar{v}_i\bar{\varepsilon}_i. \end{cases} \quad (34)$$

The approximate closed-form solution can be obtained:

$$K_i \approx 4\cdot\frac{S_{v_i\varepsilon_i}}{S_{v_iv_i}}, \quad B_i \approx 4\left(\bar{\varepsilon}_i - \frac{1}{2}\right) - 4K_i\bar{v}_i. \quad (35)$$

## A.4 LLM Usage

We only used the large language model for text polishing, solely to refine linguistic expression (e.g., sentence structure, fluency, terminological consistency) while strictly preserving the original scientific content and conclusions. No other research steps involved any large language models.

