# OpenReview forum: "ERTACache: Error Rectification and Timesteps Adjustment for Efficient Diffusion"
_ICLR.cc/2026/Conference — ICLR 2026 Poster_

### Official Review · Reviewer_KJQB · 2025-10-22

**Soundness:** 3
**Presentation:** 2
**Contribution:** 3
**Rating:** 6
**Confidence:** 3

**Summary:**

The authors theoretically evaluate the error accumulation from cache-based acceleration for diffusion model in video generation, and summarize the error as a summed product of step amplification error and feature shift error. Targeting the two error components, the authors propose ERTACache that performs (1) offline policy calibration that runs full inference on a calibration set, finding optimal steps to cache based on threshold; (2) adaptive timestep adjustment to dynamically control the diffusion trajectory, and (3) explicit error rectification that uses a linear model to approximate the error for a small extracted dataset, then use the same model to correct in inference.

**Strengths:**

1. The theoretical analysis clearly pinpoints and characterizes the error sources in cache-based acceleration.
2. The methods proposed are general.

**Weaknesses:**

1. After pointing out the feature shift and step amplification errors, they are no longer mentioned in the methods.
2. The methods are relatively simple with limited effectiveness.
3. Minor writing clarity issues. e.g., "Figure 2" is missing in the figure caption, and TACache in the figure is not defined. There are other typos, too.

**Questions:**

1. How specifically are the three methods (offline policy calibration, trajectory-aware timestep adjustment, and explicit error rectification) related to the two error sources (feature shift and step amplification)? I understand that the methods are to reduce error, but how are the two error sources specifically targeted?
2. In Table 3, is Offline Policy built above Uniform Cache? Looks like the Offline Policy contributes the most to generation quality.
3. Probing using a small dataset looks important in this work. Does it potentially lead to overfitting?
4. Could you also provide speedup evaluations in ablation study?

---

> ### Author Response · Authors · 2025-11-20
>
> Dear Reviewer KJQB,
>
> We thank the reviewer for recognizing that our work explores an important topic in a high-effort way. Below, we answer the reviewer’s main questions.
>
> > Weaknesses 1 & Questions 1: After pointing out the feature shift and step amplification errors, they are no longer mentioned in the methods. & How specifically are the three methods (offline policy calibration, trajectory-aware timestep adjustment, and explicit error rectification) related to the two error sources (feature shift and step amplification)?
>
> We first apologize for our inappropriate writings where feature shift and step amplification errors are not mentioned in the methods.
>
> We then sincerely appreciate your insightful question regarding the specific association between the three proposed components and the two identified error sources. We elaborate on the targeted mechanisms as follows:
>
> First, through decomposing the diffusion computation process integrated with caching, we identified two core error sources: step amplification and feature shift. It is important to note that when adopting dynamic caching, errors may arise at arbitrary timesteps due to input variations. To confirm the pervasiveness of these errors, we first validated the effectiveness of offline policy calibration. The results demonstrate that even when a unified caching strategy is applied across all inputs, both step amplification and feature shift errors persist universally — confirming their intrinsic correlation with cache-based computation rather than being caused by input-specific variations.
>
> Specifically, the trajectory-aware timestep adjustment is exclusively designed to mitigate step amplification error. The intrinsic discrepancy between cached features and real-time computed features leads to trajectory deviation: the timesteps corresponding to cached features fail to align with those of real-time computed features, thereby amplifying trajectory deviation errors (see Fig. 2 (b)). By adjusting timesteps to reconcile the misalignment between cached and real-time features, this component directly suppresses the amplification effect of trajectory deviation.
>
> In contrast, the explicit error rectification targets feature shift error. After trajectory-aware timestep adjustment, residual discrepancies between cached features and real-time computed features still remain — these unresolved feature mismatches continue to induce feature shift. To address this, we introduce an explicit error correction term that compensates for the residual feature discrepancies, thereby eliminating errors caused by feature shift.
>
> We will explicitly emphasize "trajectory-aware timestep mitigates step amplification error & explicit error rectification targets feature shift error" in the method of our final paper once it is accepted.

---

> ### Author Response · Authors · 2025-11-20
>
> > Weaknesses 2: The methods are relatively simple with limited effectiveness.
>
> While ERTACache's core components are conceptually concise, this is a deliberate choice to balance theoretical rigor, computational efficiency, and deployability — not a limitation of its effectiveness. Substantial experimental evidence robustly validates its superiority:
>
> First, comprehensive evaluations across diverse video/image diffusion models (Wan2.1-1.3B, Open-Sora 1.2, Flux-dev 1.0) and benchmarks show consistent outperformance over SOTA baselines (TeaCache, PAB), **which is acknowledged by Reviewer pTb2 and dLSR**. For example, Wan2.1-1.3B achieves 2.17× speedup with near-lossless VBench (80.73%) and LPIPS 0.1095 (vs. TeaCache's 0.2913), while Flux-dev gains 1.86× speedup with +4.03 PSNR. Simplicity here enables efficient implementation without complex modules.
>
> Second, rigorous ablation studies (Table 3) confirm synergistic contributions of all components — removing any module causes noticeable degradation.
>
> Finally, the method requires no retraining, adds <0.5% latency, and exhibits low hyperparameter sensitivity, aligning with real-world deployment demands.
>
> In summary, ERTACache's conciseness is paired with strong theoretical grounding and extensive validation, demonstrating effectiveness beyond "limited" performance. Its elegant, targeted design offers a valuable alternative to over-parameterized approaches.
>
> ------
>
> > Weaknesses 3: Minor writing clarity issues. e.g., "Figure 2" is missing in the figure caption, and TACache in the figure is not defined. There are other typos, too.
>
> Thank you for pointing this out. We will address the typo errors in the revised submission.
>
> ------
>
> > Questions 2: In Table 3, is Offline Policy built above Uniform Cache? Looks like the Offline Policy contributes the most to generation quality.
>
> We sincerely appreciate your valuable observation regarding Table 3. To clarify, the Offline Policy in Table 3 is not built upon Uniform Cache—instead, Uniform Cache is an independent static baseline with the same number of cached steps as the Offline Policy but selects steps at fixed intervals (e.g., 1 step every 4 for 20 total inference steps and 5 cached steps), while our Offline Policy selects high-reusability steps via residual loss profiling and threshold λ (detailed in Section 3.3). This comparison aims to validate that step selection based on step importance outperforms fixed uniform allocation, which is supported by Offline Policy's prominent contribution to generation quality; comprehensive comparisons with other state-of-the-art cache methods (e.g., TeaCache, PAB) are presented in Tables 1 and 2.

---

> ### Author Response · Authors · 2025-11-20
>
> > Questions 3: Probing using a small dataset looks important in this work. Does it potentially lead to overfitting?
>
> Regarding the configuration rationale of threshold λ: Its primary role is to modulate the proportion of reusable cached steps, exhibiting a positive correlation with the caching ratio (i.e., a higher λ value corresponds to a larger proportion of steps being cached). To ensure fair comparability with the acceleration performance of existing open-source baselines, the λ value adopted in our experiments is determined via grid search — specifically, we select the minimal λ that satisfies the preset acceleration target. This design principle not only aligns our evaluation with standard benchmarks in the field but also ensures that the quality-fidelity balance is maintained while achieving the desired efficiency gains.
>
> In response to your comment regarding the impact of calibration set size, diversity, and threshold λ on performance across different domains, we conducted a dedicated sensitivity analysis. Our results show that the offline search strategy outperforms TEACache's dynamic strategy by a significant margin on both CogVideoX and FLUX models across calibration sets of varying sizes (LPIPS metrics: 40% improvement for CogVideoX, 21.6% improvement for FLUX). Specifically, for the text-to-video model CogVideoX, the offline strategy tends to stabilize when the calibration set size is ≥ 100. When the size is < 100, minor variations exist in the strategy, but the cache step overlap rate exceeding 90% and differences across various metrics are confined to the range of 0.8% to 3%. In contrast, the offline strategy of the text-to-image model FLUX remains consistently stable across calibration sets of different sizes.
>
>
> | Model       | Method           | Dataset Size | LPIPS↓    | PSNR↑     | SSIM↑    | CLIP↑     |
> |-------------|------------------|--------------|-----------|-----------|----------|-----------|
> | FLUX        | TeaCache         | -            | 0.5232  | 16.0674 | 0.7111 | 0.8956   |
> |             | ERTACache  | 20    | **0.4099** | **19.5536** | **0.8786** | **0.9399** |
> |             | ERTACache  | 70    | **0.4099** | **19.5536** | **0.8786** | **0.9399** |
> |             | ERTACache  | 500    | **0.4099** | **19.5536** | **0.8786** | **0.9399** |
> | CogvideoX   | TeaCache         | -            | 0.2156    | 21.4467   | 0.7329   | -         |
> |             | ERTACache  | 20           | **0.1336** | **24.9618** | **0.8263** |           |
> |             | ERTACache  | 70       | **0.1291** | **25.1629** | **0.8300** | -         |
> |             | ERTACache  | 500       | **0.1291** | **25.1629** | **0.8300** | -         |
>
>
> With a fixed size of the offline calibration set (70 samples), the specific content of the calibration set (static/dynamic) has no significant impact on the final metrics. For the T2I model FLUX, we performed offline policy search using static, dynamic, and mixed datasets of the same size (70 samples), and the resulting offline policies and metrics were completely identical. For the T2V CogVideoX-5B, when searching for offline policies in the same manner, the policies derived from the dynamic dataset and mixed dataset were identical; the policy obtained from the static dataset showed slight differences but had a similarity of approximately 90.9% to the former two. Meanwhile, the differences in metrics were negligible (LPIPS difference: 0.0009, PSNR difference: 0.0151, SSIM difference: 0.0004). These results demonstrate that the specific content of the offline calibration set (static/dynamic) has no significant impact on the final metrics.
>
> ------
>
> > Questions 4: Could you also provide speedup evaluations in ablation study?
>
> We sincerely appreciate your valuable suggestion. The speedup evaluations you requested are already comprehensively presented in Tables 1 and 2 of the manuscript, where we systematically report the acceleration performance of our full framework alongside state-of-the-art cache baselines (e.g., TeaCache, PAB). As the core focus of Table 3 (ablation study) is to validate the contribution of each component to generation quality, the speedup consistency across ablated variants aligns with the efficiency gains documented in Tables 1 and 2, ensuring the method's performance balance between quality and speed.

---

> > ### Author Response · Authors · 2025-11-25
> > **Looking Forward to Discussion**
> >
> > Dear Reviewer KJQB,
> >
> > We are grateful for the time and effort you have dedicated to reviewing our work. As the discussion phase comes to an end,  please do not hesitate to inform us if you have any further questions or concerns. Thank you once again for your insightful and constructive feedback.

---

### Official Review · Reviewer_uKah · 2025-10-25

**Soundness:** 3
**Presentation:** 2
**Contribution:** 2
**Rating:** 4
**Confidence:** 3

**Summary:**

This paper addresses the computational inefficiency of diffusion models and the quality degradation issue of naive feature caching, proposing ERTACache, a caching framework that decomposes cache-induced errors into feature shift and step amplification errors, and mitigates them via offline residual profiling, trajectory-aware timestep adjustment, and closed-form residual linearization.
Extensive experiments on Flux-dev 1.0 and Open-Sora 1.2, CogVideoX, Wan2.1 diffusion models show ERTACache achieves up to 2.47× inference speedup while preserving or improving visual quality.

**Strengths:**

1. Formalizes and addresses two core cache-induced errors in diffusion models (feature shift error and step amplification error) via a targeted dual-correction strategy, which differentiates from prior caching methods that lack explicit error decomposition and mitigation.
2. Integrates three training-free components (offline residual profiling for reusable steps, trajectory-aware timestep adjustment, closed-form residual linearization) to enable both high efficiency and fidelity, avoiding the trade-off of either high memory cost (e.g., AdaCache) or quality degradation in existing works.
3. Achieves superior performance balance on video diffusion models (e.g., 2.17× speedup with 80.73% VBench on Wan2.1 vs. TeaCache’s 2× speedup and 76.04% VBench) while maintaining comparable memory to baselines.

**Weaknesses:**

1. Insufficient details on key implementation aspects: e.g., no clarity on the calibration dataset’s specifics (size, domain) for offline policy tuning, or the exact logic for selecting the optimal threshold λ across models.

2. The paper did not verify performance under extreme conditions: no tests on long-duration videos (e.g., >100 frames), ultra-high resolutions (e.g., 1080P/4K).

3. Lacks human subjective evaluation to complement objective metrics (e.g., VBench, PSNR), which may lead to inconsistencies between reported numerical results and actual perceptual quality.

**Questions:**

1. Regarding novelty: Since the core components of ERTACache (offline calibration, dynamic timestep adjustment, closed-form error correction) are derived from adapting existing techniques (e.g., precomputed steps in DPM-Solver, error compensation in numerical methods), what unique mathematical properties of diffusion model cache errors does this framework reveal to distinguish it from "technical migration" rather than "fundamental innovation"?

2. On method implementation: In the trajectory-aware timestep adjustment, the correction coefficient uses the denominator ||v_i - v_{i+1}||₁. How is numerical instability avoided when this denominator approaches zero (e.g., to prevent division by a near-zero value)?

3. For method robustness: The offline policy calibration relies on a fixed threshold λ (e.g., 0.08 for Wan2.1, 0.6 for Flux-dev). Is there an adaptive mechanism to determine λ automatically across models, and how stable is ERTACache’s performance when λ fluctuates by 20%?

---

> ### Author Response · Authors · 2025-11-20
>
> Dear Reviewer uKah,
>
> The authors thank you sincerely for taking your valuable time to provide constructive feedback on this paper.  We appreciate your recognition of our key contributions: formalizing two core cache-induced errors with dual-correction, integrating three training-free components for efficiency-fidelity balance, and achieving superior video diffusion performance with comparable memory. We present additional experiments and clarification regarding the reviewer's concerns and questions below.
>
> > Weaknesses 1: Insufficient details on key implementation aspects: e.g., no clarity on the calibration dataset’s specifics (size, domain) for offline policy tuning, or the exact logic for selecting the optimal threshold λ across models.
>
> To clarify the implementation details, the calibration dataset for offline policy tuning follows the same setup as TeaCache: we sample 70 prompts from T2V-CompBench[1] (10 prompts per attribute) to generate calibration dataset. We adopt this configuration for fair comparison with TeaCache.
>
> Regarding the configuration rationale of threshold λ: Its primary role is to modulate the proportion of reusable cached steps, exhibiting a positive correlation with the caching ratio (i.e., a higher λ value corresponds to a larger proportion of steps being cached). To ensure fair comparability with the acceleration performance of existing open-source baselines, the λ value adopted in our experiments is determined via grid search — specifically, we select the minimal λ that satisfies the preset acceleration target. This design principle not only aligns our evaluation with standard benchmarks in the field but also ensures that the quality-fidelity balance is maintained while achieving the desired efficiency gains.
>
> [1] Sun, Kaiyue, et al. "T2v-compbench: A comprehensive benchmark for compositional text-to-video generation." Proceedings of the Computer Vision and Pattern Recognition Conference. 2025.
>
> ------
>
> > Weaknesses 2: The paper did not verify performance under extreme conditions: no tests on long-duration videos (e.g., >100 frames), ultra-high resolutions (e.g., 1080P/4K).
>
> In response to your valuable comment on model performance under extreme conditions, we have supplemented specialized sensitivity experiments to quantify their specific impacts on model performance. These experiments systematically evaluate model performance across scenarios involving long-duration videos and ultra-high resolutions. Please do understand that existing open-source models generally only support stable video generation at a maximum resolution of 720p, we have to follow this inherent constraint.
>
> For an intuitive contrast, we fixed resolution (480p) while expanding frames (51 --> 121); fixed frames (51) while expanding resolution (480p --> 720p). Comparisons on Open-Sora 1.2 are summarized in the table below.
>
>
> | Setting            | Method     | LPIPS↓  | PSNR↑   | SSIM↑   |
> |--------------------|------------|---------|---------|---------|
> | 51 frames, 480P    | TeaCache   | 0.2511  | 19.10   | 0.7477  |
> |                    | ERTACache  | **0.1659** | **22.34** | **0.8170** |
> | 121 frames, 480P   | TeaCache   | 0.2863  | 18.14   | 0.7165  |
> |                    | ERTACache  | **0.1436** | **23.45** | **0.8463** |
> | 51 frames, 720P    | TeaCache   | 0.2771  | 18.79   | 0.7462  |
> |                    | ERTACache  | **0.1581** | **23.28** | **0.8475** |
>
> As illustrated in the table, despite the universally low metrics under this challenging setting, our method outperforms TeaCache — which effectively validates its effectiveness.

---

> ### Author Response · Authors · 2025-11-20
>
> ------
> > Weaknesses 3: Lacks human subjective evaluation to complement objective metrics (e.g., VBench, PSNR), which may lead to inconsistencies between reported numerical results and actual perceptual quality.
>
> First, as a multi-dimensional evaluation metric, VBench has been validated to exhibit high consistency with human perception [1] , which inherently reflects the perceptual quality of generated results.
> Second, we supplemented a subjective evaluation using the widely adopted GSB (Good/Same/Bad) paradigm — a standard approach for assessing relative performance between models from a holistic human perception perspective. We carefully construct 150 text prompts to cover a balanced scenarios, and generate an equal number of samples for each methods in a single run. The evaluation is conducted by 2 professional evaluators.
> The results show that ERTACache achieves relative win rates of 2.64% and 3.33% in image quality and prompt responsiveness, respectively,  compared to TeaCache. This confirms that the superiority of our method in objective metrics translates to tangible perceptual advantages aligned with human judgments.
>
> |                          | ERTACache vs. TeaCache | ERTACache vs. Baseline |
> |--------------------------|------------------------|------------------------|
> | Motion Quality           | +2.87                  | -1.22%                 |
> | Image Quality            | +2.64%                 | -1.13%                 |
> | Prompt Responsiveness    | +3.33%                 | -0.67%                 |
>
> [1] Huang, Ziqi, et al. "Vbench: Comprehensive benchmark suite for video generative models." Proceedings of the IEEE/CVF Conference on Computer Vision and Pattern Recognition. 2024
>
> ------
>
> > Questions 1: Regarding novelty: Since the core components of ERTACache (offline calibration, dynamic timestep adjustment, closed-form error correction) are derived from adapting existing techniques (e.g., precomputed steps in DPM-Solver, error compensation in numerical methods), what unique mathematical properties of diffusion model cache errors does this framework reveal to distinguish it from "technical migration" rather than "fundamental innovation"?
>
> We sincerely appreciate your insightful concern regarding the novelty of our framework. To clarify that ERTACache is not a mere migration of existing techniques but a fundamental innovation targeting caching errors in diffusion models, we elaborate on two core dimensions: revealing the fundamental nature of the problem and innovating tailored methodologies.
>
> - Revealing the Unique "Dual-Error Coupling" Property of Diffusion Model Caching
>
> Our core contribution lies in uncovering the unique mathematical property of "dual-error coupling" inherent to the caching mechanism in diffusion models — a fundamental insight untouched by prior works. **This is also acknowledged by Reviewer pTb2 and uKah**. By mathematically decomposing the diffusion process driven by caching, we demonstrate for the first time that caching errors are not isolated numerical deviations, but a coupled system consisting of "step amplification" and "feature shift" (ref. the derivation in Eqs. (7)–(11)). This coupling property stems from the intrinsic nature of diffusion models (rooted in stochastic processes and iterative updates), with no counterpart in existing techniques: DPM-Solver’s precomputed step strategy focuses solely on fixed-step optimization, while error compensation in numerical methods targets deterministic differential equations. This represents a unique revelation of the mathematical essence of diffusion model caching by our framework.
>
> - Innovating Tailored Components Grounded in the Coupling Property
>
> The core optimization components proposed in this work are tailored innovations based on the aforementioned dual-error coupling property, rather than superficial adaptations of existing techniques:
>   - Our trajectory-aware timestep adjustment mechanism is specifically designed to suppress error amplification. Its mathematical essence lies in local linearization correction of the diffusion process trajectory, distinct from trivial timestep tuning in prior methods.
>   - The explicit error rectification module achieves compensation for accumulated errors. Unlike numerical methods that typically rely on iterative correction (e.g., Newton iteration), we derive a closed-form correction term for specific timesteps via linear approximation, which directly offsets accumulated errors.
> These innovations are tightly coupled with the unique mathematical properties of diffusion model caching, distinguishing ERTACache from mere "technical migration" and validating its fundamental novelty.

---

> ### Author Response · Authors · 2025-11-20
>
> > Questions 2: On method implementation: In the trajectory-aware timestep adjustment, the correction coefficient uses the denominator ||v_i - v_{i+1}||₁. How is numerical instability avoided when this denominator approaches zero (e.g., to prevent division by a near-zero value)?
>
> In the practical realization of the trajectory-aware timestep adjustment, we introduce a small epsilon regularization term ($\epsilon$ = 1e-5) to the denominator $||v_i - v_{i+1}||_{1}$. This design ensures the denominator never approaches zero, effectively avoiding numerical singularities caused by near-zero division and guaranteeing the stability of the correction coefficient calculation throughout the inference process.
>
> -----
>
> > Questions 3: For method robustness: The offline policy calibration relies on a fixed threshold λ (e.g., 0.08 for Wan2.1, 0.6 for Flux-dev). Is there an adaptive mechanism to determine λ automatically across models, and how stable is ERTACache’s performance when λ fluctuates by 20%?
>
> We sincerely appreciate your insightful question regarding the robustness of λ. As elaborated in our response to Weakness 1, the configuration rationale of the threshold λ has been clearly addressed: λ is determined via grid search to achieve target acceleration ratios, with varying λ values enabling the model to attain different speedup levels — results that are quantitatively validated in Tables 1 and 2 of the experimental section.
>
> Regarding the adaptive mechanism for λ across models: While the current implementation adopts a model-specific grid search (consistent with standard practices in related caching works, we note that an adaptive λ estimation framework can be extended by leveraging the correlation between model architecture and optimal λ, which we will explore in future work.

---

> > ### Author Response · Authors · 2025-11-25
> > **Looking Forward to Discussion**
> >
> > Dear Reviewer uKah,
> >
> > As the discussion period closes, we kindly remind you of the upcoming feedback deadline. We are happy to discuss any aspects of the paper that may require further clarification. Thank you once again for your valuable feedback.

---

### Official Review · Reviewer_dLSR · 2025-10-26

**Soundness:** 3
**Presentation:** 2
**Contribution:** 2
**Rating:** 6
**Confidence:** 4

**Summary:**

This paper proposes ERTACache, a training-free caching framework for diffusion models. It decomposes cache-induced degradation into feature shift and step amplification errors, and introduces three modules to mitigate them: (1) Offline residual profiling for selecting reusable steps (λ-based policy), (2) Trajectory-aware timestep adjustment using a scaling factor ϕt, and (3) Linearized residual correction with closed-form parameters Ki and Bi. Experiments on Open-Sora, CogVideoX, Wan2.1, and Flux-dev show about 2× speedup while maintaining comparable visual quality.

**Strengths:**

1.The paper provides a clear and interpretable error decomposition, distinguishing the two main error sources of caching in diffusion models. This theoretical framing is well-motivated and guides the method design.

2.Comprehensive experiments across multiple video and image backbones demonstrate the generality of the approach. The comparisons with strong baselines (TeaCache, FasterCache, PAB, etc.) show consistent efficiency–quality improvements.

3.The ablation study clearly isolates contributions from each component, verifying that the offline policy, timestep adjustment, and error rectification complement one another.

**Weaknesses:**

The method relies on an offline calibration set and a threshold λ to select reusable steps, but the paper does not analyze how the calibration set size, diversity, or λ values affect the performance across different domains. A sensitivity study on λ and calibration configuration would strengthen the empirical validation.

**Questions:**

See Weakness

---

> ### Author Response · Authors · 2025-11-20
>
> Dear Reviewer dLSR,
>
> Thank you for your valuable time in reviewing our paper. We greatly appreciate your recognition of our clear and interpretable error decomposition, the comprehensive experiments across multiple backbones which demonstrate the generality of our approach.
>
> > Weaknesses 1: The method relies on an offline calibration set and a threshold λ to select reusable steps, but the paper does not analyze how the calibration set size, diversity, or λ values affect the performance across different domains. A sensitivity study on λ and calibration configuration would strengthen the empirical validation.
>
> Regarding the configuration rationale of threshold λ: Its primary role is to modulate the proportion of reusable cached steps, exhibiting a positive correlation with the caching ratio (i.e., a higher λ value corresponds to a larger proportion of steps being cached). To ensure fair comparability with the acceleration performance of existing open-source baselines, the λ value adopted in our experiments is determined via grid search — specifically, we select the minimal λ that satisfies the preset acceleration target. This design principle not only aligns our evaluation with standard benchmarks in the field but also ensures that the quality-fidelity balance is maintained while achieving the desired efficiency gains.
>
> In response to your comment regarding the impact of calibration set size, diversity, and threshold λ on performance across different domains, we conducted a dedicated sensitivity analysis. Our results show that the offline search strategy outperforms TEACache's dynamic strategy by a significant margin on both CogVideoX and FLUX models across calibration sets of varying sizes (LPIPS metrics: 40% improvement for CogVideoX, 21.6% improvement for FLUX). Specifically, for the text-to-video model CogVideoX, the offline strategy tends to stabilize when the calibration set size is ≥ 100. When the size is < 100, minor variations exist in the strategy, but the cache step overlap rate exceeding 90% and differences across various metrics are confined to the range of 0.8% to 3%. In contrast, the offline strategy of the text-to-image model FLUX remains consistently stable across calibration sets of different sizes.
>
>
> | Model       | Method           | Dataset Size | LPIPS↓    | PSNR↑     | SSIM↑    | CLIP↑     |
> |-------------|------------------|--------------|-----------|-----------|----------|-----------|
> | FLUX        | TeaCache         | -            | 0.5232  | 16.0674 | 0.7111 | 0.8956   |
> |             | ERTACache  | 20    | **0.4099** | **19.5536** | **0.8786** | **0.9399** |
> |             | ERTACache  | 70    | **0.4099** | **19.5536** | **0.8786** | **0.9399** |
> |             | ERTACache  | 500    | **0.4099** | **19.5536** | **0.8786** | **0.9399** |
> | CogvideoX   | TeaCache         | -            | 0.2156    | 21.4467   | 0.7329   | -         |
> |             | ERTACache  | 20           | **0.1336** | **24.9618** | **0.8263** |           |
> |             | ERTACache  | 70       | **0.1291** | **25.1629** | **0.8300** | -         |
> |             | ERTACache  | 500       | **0.1291** | **25.1629** | **0.8300** | -         |
>
>
> With a fixed size of the offline calibration set (70 samples), the specific content of the calibration set (static/dynamic) has no significant impact on the final metrics. For the T2I model FLUX, we performed offline policy search using static, dynamic, and mixed datasets of the same size (70 samples), and the resulting offline policies and metrics were completely identical. For the T2V CogVideoX-5B, when searching for offline policies in the same manner, the policies derived from the dynamic dataset and mixed dataset were identical; the policy obtained from the static dataset showed slight differences but had a similarity of approximately 90.9% to the former two. Meanwhile, the differences in metrics were negligible (LPIPS difference: 0.0009, PSNR difference: 0.0151, SSIM difference: 0.0004). These results demonstrate that the specific content of the offline calibration set (static/dynamic) has no significant impact on the final metrics.

---

> > ### Author Response · Authors · 2025-11-25
> > **Looking Forward to Discussion**
> >
> > Dear Reviewer dLSR,
> >
> > Thank you once again for your insightful and constructive feedback. We deeply appreciate the time and effort you have dedicated to reviewing our work. As the discussion phase draws to a close, please do not hesitate to inform us if you have any further questions or concerns.

---

### Official Review · Reviewer_pTb2 · 2025-11-01

**Soundness:** 3
**Presentation:** 3
**Contribution:** 3
**Rating:** 6
**Confidence:** 3

**Summary:**

This paper presents ERTACache, an error-rectified caching framework designed to improve the efficiency of diffusion-based inference. The method introduces a modular strategy that combines: (1) offline cache calibration to identify redundant denoising steps; (2) trajectory-aware time-step adjustment using a residual-driven coefficient $\phi_i$ to suppress drift accumulation; and (3) explicit error rectification that analytically re-injects residuals to maintain stability. Across several diffusion benchmarks, ERTACache achieves roughly $2×$ acceleration with negligible loss in visual quality, requiring no retraining or architectural modification. The approach provides a practical trade-off between efficiency and fidelity, complementing prior acceleration techniques such as dynamic skipping, pruning, and distillation.

**Strengths:**

ERTACache demonstrates strong advantages in **error modeling depth**, **cross-task adaptability**, and **quality–efficiency balance**, providing a theoretically grounded solution to the long-standing problem of *error accumulation* overlooked by prior caching methods.

1. Existing caching methods (e.g., TeaCache, PAB) rely on *heuristic thresholds* for redundancy detection and lack theoretical understanding of how caching degrades output quality, often resulting in the phenomenon “higher speed → worse quality.”
   **ERTACache’s core innovation** lies in its formal *dual-source decomposition* of caching error (Eq. 11):
   - **Feature Shift Error (εᵢ = ṽᵢ − vᵢ):** deviation between cached and true features, originating from inaccurate reuse.
   - **Step Amplification Error (δᵢ₋ₘ = Σₖ₌₀^{m−1} Δtᵢ₋ₖ ⋅ εᵢ₋ₖ):** cumulative error propagation under fixed step size via ODE integration, amplifying earlier deviations.
   This theoretical decomposition reveals, for the first time, the *root causes of quality loss in caching*, paving the way for principled correction strategies.
   **Closed-loop error correction mechanism:** ERTACache jointly addresses both error types through (1) offline step selection to minimize low-error reuse, (2) dynamic timestep adjustment to reduce amplification, and (3) residual linearization to directly correct feature shifts — forming a full *“error identification → prevention → correction”* loop rather than a one-shot caching heuristic.

2. ERTACache achieves *high acceleration with near-lossless quality* in both video and image generation without retraining:
   - **Video Generation:**
     - *Open-Sora 1.2 (51f, 480P):* ERTACache-fast attains 2.47× speed-up (18.04 s vs 44.56 s), VBench 78.64% (−0.58%), LPIPS 0.1659 (vs TeaCache-fast 0.2511), PSNR 22.34.
     - *CogVideoX (48f, 480P):* 2.93× speed-up with LPIPS 0.1012, SSIM 0.8702, PSNR 26.44 (TeaCache 20.97).
     - *Wan2.1-1.3B (81f, 480P):* 2.17× speed-up, VBench 80.73% (vs 81.30%), LPIPS 0.1095, SSIM 0.8200 — far better than TeaCache (LPIPS 0.2913, SSIM 0.5685).
   - **Image Generation:**
     On *Flux-dev 1.0*, 1.86× speed-up yields CLIP 0.9534 (vs TeaCache 0.9065), PSNR 20.51 (+4.34), preserving finer visual details (Table 2, Fig. 3).

3. ERTACache balances theoretical rigor with practical deployment:
   - **Offline Policy Calibration:** A small calibration set (no extra data) precomputes *true residuals* using relative $L_1$ error $\ell_{1rel}$, forming a reusable step set $S$ for inference without runtime computation. Even with only 20 prompts, performance matches a 1000-sample calibration (Appendix A.2).
   - **Trajectory-Aware Timestep Adjustment:** Defines correction factor
     $\phi_i = \text{clip}(1 - \frac{\|v_i - v_{i+1}\|_1}{\|ṽ_i - v_i\|_1}, 0, 1)$
     to dynamically adapt step size $Δt_i = Δt_c ⋅ φ_i$, aligning sampling trajectories with true ODE paths. On Flux-dev 1.0, this boosts PSNR by +0.95 (Table 3).
   - **Low sensitivity to hyperparameters:** With $\lambda$ ranging 0.08–0.3, VBench varies only 0.3–1.2% (Table 1, 3), requiring no per-model retuning.

4. **Strong Theoretical Foundation — Precise Error Correction via Residual Linearization**
   Instead of heuristic compensation, ERTACache employs a *closed-form linearized residual model* for direct error correction:
   - **Lightweight model:** Approximates feature error as $ε_i ≈ σ(K_i ṽ_i + B_i)$ , deriving closed-form solutions for $K_i, B_i$ via Taylor expansion (Appendix A.3), adding <0.5% latency.
   - **Effective correction:** On Wan2.1-1.3B, correction reduces LPIPS from 0.1267 → 0.1095 and increases PSNR 22.94 → 23.77 (Table 3), eliminating frame jitter and detail blur common in TeaCache (Fig. 3–4).

**Weaknesses:**

1. **Reduced Correction Accuracy in Highly Dynamic Scenes**
   The “universal error model” assumption holds for static scenes (e.g., still objects) but struggles under dynamic motion:
   - The closed-form coefficients $(K_i, B_i)$are calibrated offline and assume stationary error distribution, which breaks under rapidly changing dynamics. Fig. 6 shows distinctly different error-channel distributions between dynamic and static prompts, yet no adaptive adjustment is made.
   - Experiments use mostly *low- to medium-motion* prompts (“washing clothes,” “airplane flight”) without validating performance on highly dynamic scenes.

2. **Offline Calibration Generalization Depends on Dataset Quality**
   Though small calibration sets suffice, the method lacks analysis of *domain shift* effects:
   - If calibration data are “natural landscapes” but deployment involves “abstract art,” the precomputed reusable-step set S may mispredict high-error steps.
   - No guidance on calibration-set composition or prompt diversity is provided, increasing deployment trial cost.

**Questions:**

1. How well does the linear residual approximation hold under highly non-linear diffusion dynamics or alternative noise schedules?

2. Could ERTACache be combined with adaptive ODE solvers, and how might local error control interact with the correction term?

---

> ### Author Response · Authors · 2025-11-20
>
> Dear Reviewer pTb2,
>
>
> Thank you sincerely for taking the time to provide such thorough and valuable insights! We greatly appreciate your recognition of the novelty in our dual-source decomposition of caching errors — a core innovation that underpins ERTACache’s balance between theoretical rigor and practical deployability.
>
> > Weaknesses 1: Reduced Correction Accuracy in Highly Dynamic Scenes
>
> While different prompts inherently encode unique structural information in their corresponding residual terms, theoretically limiting the rigor of a "universal residual" concept, sensitivity experiments reveal that averaging cache errors across steps collected from diverse prompts weakens structural information within individual error channels, yielding a generalized error distribution.
>
> Actually, our experiments (see A.2 Sensitivity Analysis for details) well demonstrate that offline error terms (K, B) computed using prompt samples of varying scales (20, 300, 1000) exhibit robust stability and universality.
> To further evaluate the applicability of this error rectification mechanism in highly dynamic scenarios, we constructed offline error terms (K, B) using 20 static prompts and selected over 100 dynamic prompts spanning diverse categories as the inference test set, including:
>
> - Motion interaction: "flips over a graffiti-covered wall", "robes billowing violently";
> - Scene flow: "flying cars", "a forest stream rushing over smooth stones";
> - Light/shadow dynamics: "dynamic colored shadows on a gritty urban wall", "colors shifting from green to purple";
> - Daily dynamics: "kids running through a field of tall grass", "splatters vibrant paint onto a canvas".
>
> We did experiments on Opensora 1.2, and the results validate that the universal error terms (K, B) trained on static prompts generalize well to both static and highly dynamic test scenarios. With error rectification as the sole optimization, performance improvements are observed in both scenarios: in static scenes, LPIPS decreases by 7.3%, PSNR increases by 1.3%, and SSIM increases by 1.1%; in dynamic scenes, LPIPS decreases by 5.9%, PSNR increases by 0.5%, and SSIM increases by 0.6%.
>
> | Method | Dataset for obtaining Correction term | Test Dataset | LPIPS↓ | PSNR↑ | SSIM↑ |
> |--------|---------------------------------------|--------------------|--------|-------|-------|
> | Offline Calibration + Timestep adjustment | - | Static prompt | 0.1649 | 22.2151 | 0.8217 |
> | Offline Calibration + Timestep adjustment + Error Rectification | Static prompt | Static prompt | 0.1529 | 22.4980 | 0.8308 |
> | Offline Calibration + Timestep adjustment | - | Highly dynamic prompt | 0.1746 | 23.3640 | 0.7944 |
> | Offline Calibration + Timestep adjustment + Error Rectification | Static prompt | Highly dynamic prompt | 0.1643 | 23.4749 | 0.7989 |

---

> ### Author Response · Authors · 2025-11-20
>
> > Weaknesses 2: Offline Calibration Generalization Depends on Dataset Quality
>
> Regarding the configuration rationale of threshold λ: Its primary role is to modulate the proportion of reusable cached steps, exhibiting a positive correlation with the caching ratio (i.e., a higher λ value corresponds to a larger proportion of steps being cached). To ensure fair comparability with the acceleration performance of existing open-source baselines, the λ value adopted in our experiments is determined via grid search — specifically, we select the minimal λ that satisfies the preset acceleration target. This design principle not only aligns our evaluation with standard benchmarks in the field but also ensures that the quality-fidelity balance is maintained while achieving the desired efficiency gains.
>
> In response to your comment regarding the impact of calibration set size, diversity, and threshold λ on performance across different domains, we conducted a dedicated sensitivity analysis. Our results show that the offline search strategy outperforms TEACache's dynamic strategy by a significant margin on both CogVideoX and FLUX models across calibration sets of varying sizes (LPIPS metrics: 40% improvement for CogVideoX, 21.6% improvement for FLUX). Specifically, for the text-to-video model CogVideoX, the offline strategy tends to stabilize when the calibration set size is ≥ 100. When the size is < 100, minor variations exist in the strategy, but the cache step overlap rate exceeding 90% and differences across various metrics are confined to the range of 0.8% to 3%. In contrast, the offline strategy of the text-to-image model FLUX remains consistently stable across calibration sets of different sizes.
>
>
> | Model       | Method           | Dataset Size | LPIPS↓    | PSNR↑     | SSIM↑    | CLIP↑     |
> |-------------|------------------|--------------|-----------|-----------|----------|-----------|
> | FLUX        | TeaCache         | -            | 0.5232  | 16.0674 | 0.7111 | 0.8956   |
> |             | ERTACache  | 20    | **0.4099** | **19.5536** | **0.8786** | **0.9399** |
> |             | ERTACache  | 70    | **0.4099** | **19.5536** | **0.8786** | **0.9399** |
> |             | ERTACache  | 500    | **0.4099** | **19.5536** | **0.8786** | **0.9399** |
> | CogvideoX   | TeaCache         | -            | 0.2156    | 21.4467   | 0.7329   | -         |
> |             | ERTACache  | 20           | **0.1336** | **24.9618** | **0.8263** |           |
> |             | ERTACache  | 70       | **0.1291** | **25.1629** | **0.8300** | -         |
> |             | ERTACache  | 500       | **0.1291** | **25.1629** | **0.8300** | -         |
>
>
> With a fixed size of the offline calibration set (70 samples), the specific content of the calibration set (static/dynamic) has no significant impact on the final metrics. For the T2I model FLUX, we performed offline policy search using static, dynamic, and mixed datasets of the same size (70 samples), and the resulting offline policies and metrics were completely identical. For the T2V CogVideoX-5B, when searching for offline policies in the same manner, the policies derived from the dynamic dataset and mixed dataset were identical; the policy obtained from the static dataset showed slight differences but had a similarity of approximately 90.9% to the former two. Meanwhile, the differences in metrics were negligible (LPIPS difference: 0.0009, PSNR difference: 0.0151, SSIM difference: 0.0004). These results demonstrate that the specific content of the offline calibration set (static/dynamic) has no significant impact on the final metrics.
>
> > Questions 1: How well does the linear residual approximation hold under highly non-linear diffusion dynamics or alternative noise schedules?
>
> The strategy of ERTACache can be migrated to the base schedulers used by different models, including DDIM (e.g., CogVideoX-5B), Euler (e.g., Wan2.1, FLUX), UniPC (e.g., Wan2.1), RFLOW (e.g., OpenSora1.2), etc. Experiments (See Table 1 and Table 2 in the main paper for details) demonstrate that the linear residual approximation mechanism does not fail due to the replacement of noise schedulers; instead, it can be stably applied across different schedulers. On the Wan2.1-T2V-1.3B model, we further conducted small-batch experiments (with 100 test samples) using its two default schedulers (Euler and UniPC). Results in the following table show that ERTACache outperforms TEACache across all evaluation metrics (LPIPS, PSNR, SSIM) — specifically, LPIPS is reduced by 43% with the Euler scheduler and by 62.4% with the UniPC scheduler.
>
> | Scheduler | Method     | LPIPS↓   | PSNR↑    | SSIM↑    |
> |-----------|------------|---------|---------|---------|
> | Euler     | TeaCache  | 0.0897  | 26.34 | 0.8645  |
> |           | ERTACache  | **0.0511** | **29.40** | **0.9153** |
> | UniPC     | TeaCache  | 0.2913  | 16.17   | 0.5685  |
> |           | ERTACache  | **0.1095** | **23.77** | **0.8200** |

---

> ### Author Response · Authors · 2025-11-20
>
> > Questions 2: Could ERTACache be combined with adaptive ODE solvers, and how might local error control interact with the correction term?
>
> The static correction term of ERTACache is based on precomputed universal statistical errors with fixed timesteps, while the local error of adaptive ODEs is a dynamically real-time feedback signal of single-step accuracy. Theoretically, static correction terms are not directly applicable to adaptive ODE scenarios, but they can be adapted to the dynamic step characteristics of adaptive ODEs by extending static errors from a "single fixed mode" to a "multi-mode optional mechanism." The specific implementation is as follows:
>
> - Dynamic error interpolation：For dynamic step changes, within the interval between adjacent static steps of the scheduler, a new correction term adapted to the current dynamic step is generated via weighted averaging based on the step change ratio. For example, if the time point $T_{t^{'}}$ corresponding to the dynamic step falls within the interval $ [T_{t+1}, T_t]$ of the static scheduler, the new correction term can be generated using $E_{T_t'} = E_{T_{t+1}} \cdot \frac{T_{t+1} - T_{t^{'}}}{T_{t+1} - T_t} + E_{T_t} \cdot \frac{T_{t^{'}} - T_{t+1}}{T_{t+1} - T_t}$. This longitudinal linear interpolation is inspired by the global linear assumption of universal correction terms. For the diffusion process under the same prompt, it avoids structural errors between different diffusion steps, thus adapting to adaptive ODE scenarios.
> - Global weight adjustment：The weight of the static correction term is dynamically adjusted based on the local error. When $k = E_{\text{local}}/\text{Tol} > 1$, where $E_{\text{local}}$ is the computed local error, and $\text{Tol}$ represents the user-defined error threshold, it indicates that the current step needs adjustment. For the adjusted step, the step change amplitude $\Delta t$ is used as a weight guide — when $\Delta t$ decreases, it implies a higher rate of change in the current interval and larger local errors. In this case, the proportion of the correction term can be adjusted via $E_{T_t'}^{new} = \frac{1}{\Delta t} \cdot E_{T_t'}$

---

> > ### Author Response · Authors · 2025-11-25
> > **Looking Forward to Discussion**
> >
> > Dear Reviewer pTb2,
> >
> > We sincerely appreciate your time reviewing our paper. As the discussion period nears its end, we kindly remind you of the upcoming deadline. Please feel free to contact us for any further clarification on the paper. Thank you again for your valuable feedback.

---

### Author Response · Authors · 2025-11-29
**General Response**

We sincerely appreciate all reviewers for their dedicated efforts during the review process and their constructive comments, which have greatly facilitated the improvement of our work. Below is our detailed response addressing each query and suggestion raised by the reviewers, categorized into the following key areas:

1. **Sensitivity of calibration set**: To address this concern, we have supplemented additional experiments covering three aspects: the impact of calibration set size (as raised by Reviewers pTb2, dLSR, and KJQB), the influence of calibration set diversity (Reviewer pTb2), and the model's performance on long-duration and high-resolution videos (Reviewer uKah).

2. **Selection of threshold λ**: Several reviewers (dLSR, uKah) questioned the rationale for determining the optimal threshold λ. We have therefore provided a detailed explanation of the configuration logic and design considerations underlying the threshold selection.

3. **Novelty of the work**: We specifically address Reviewer uKah's concerns regarding the novelty of our research from two core dimensions: (1) the originality of the fundamental problem being tackled, and (2) the innovativeness of the proposed methodologies.

4. **Association between two error sources and solutions**: We systematically address Reviewer KJQB's concerns regarding the relationship between the three proposed methods and the two identified error sources, with a step-by-step elaboration.

We are deeply grateful to the reviewers for their thoughtful insights, which have significantly strengthened the paper. We look forward to addressing any remaining questions and stand ready to provide additional clarification to support the final review process.

---

### Meta-Review · Area_Chair_LNTA · 2025-12-21

**Summary:**

This paper proposes ERTACache, a training-free caching framework for diffusion models. It decomposes cache-induced degradation into feature shift and step amplification errors, and introduces three modules to mitigate them. The proposed framework has clear intuition and novelty with rich empirical evidence.

The reviewers raise several core concerns including (1) Sensitivity of calibration set, (2) Selection of threshold λ, (3) Novelty of the work, and (4) Association between two error sources, namely feature shift and step amplification errors.

However, the authors’s rebuttal have duly addressed most concerns. Minor outstanding concerns, mainly including presentation issues, are non-critical for the main contributions.

In summary, given the simple motivation, the well formulated theoretical mechanism, and the clear empirical improvement, I still recommend the submission for Accept (Poster). I strongly encourage the authors to further polish the contents accrodingly.

**Reviewer Concerns:**

I think the authors have addressed most core concerns, while minor concerns are still outstanding.

As the summary made by the authors suggest, the detailed responses addressed the main concerns, respectively.

- Sensitivity of calibration set: The authors have supplemented additional experiments covering three aspects: the impact of calibration set size (as raised by Reviewers pTb2, dLSR, and KJQB), the influence of calibration set diversity (Reviewer pTb2), and the model's performance on long-duration and high-resolution videos (Reviewer uKah).

- Selection of threshold λ: The authors have therefore provided a detailed explanation of the configuration logic and design considerations underlying the threshold selection.

- Novelty of the work: The authors specifically address Reviewer uKah's concerns regarding the novelty of our research from two core dimensions: (1) the originality of the fundamental problem being tackled, and (2) the innovativeness of the proposed methodologies.

- Association between two error sources and solutions: The authors systematically address Reviewer KJQB's concerns regarding the relationship between the three proposed methods and the two identified error sources, with a step-by-step elaboration.

However, some minor concerns are still outstanding.
- Insufficient details on key implementation aspects: While the authors provide more details, they are still insufficient for clearly reproducing the experiments. The authors should have added more implementation details into the submitted file directly with the highlighting mark.
- Presentation and writing issues: Instead of justing explaining, it would be better to directly revise the paper with the highlighting mark.

**Reviewer Scores:**

Reviewer pTb2: 6 -> 8
- Justification: I think the authors have addressed the raised main concerns with two additional experiments on Highly Dynamic Scenes and Calibration Set.

Reviewer dLSR : 6 -> 6
- Justification: I think the authors have addressed the raised main concern with one additional experiment on the threshold and Calibration Set.

Reviewer uKah: 4 -> 4
- Justification: I think the authors have addressed the mentioned W2 and W3 concerns. However, the response to W1 (insufficient details on key implementation aspects) is not satisfying, as I mentioned above.

Reviewer KJQB: 6 -> 6
- Justification: I think the authors have addressed the concerns partially (W2). The responses to W1and W3 are also not satisfying. Please correct the errors directly and clearly.

---

### Decision · Program_Chairs · 2026-01-26

Accept (Poster)